# Experiences and needs of patients facing incurable cancer and their relatives with informal care in psychosocial supporting centres in the Netherlands: A qualitative study

Suzanne D. Siemelink[1], Julia van Koeveringe[1], Monique Bussmann[2], Saskia van Veen[3], Sandrina Sangers[4], Natasja J.H. Raijmakers [1]*

1 Department of Research & Development, Netherlands Comprehensive Cancer Organisation (IKNL), Utrecht, The Netherlands, 2 Rotterdam University of Applied Sciences, Research Center Innovations in Care, Rotterdam, The Netherlands, 3 IPSO Branch Organization, Almere, The Netherlands, 4 Agora, Bunnik, The Netherlands

* n.raijmakers@iknl.nl

## Abstract

### Purpose

People facing incurable cancer need appropriate informal psychosocial care. This study examines the experiences and needs of visitors of IPSO centers (psychosocial supporting centers for life with and after cancer) for people facing incurable cancer in order to optimize the services provided by IPSO centers for this group.

### Method

Semi-structured interviews were conducted with visitors of IPSO centers, both patients and relatives facing incurable cancer. The interviews were analyzed thematically. Eight themes and two sub themes were identified.

### Results

In total, 18 patients and 13 relatives and bereaved relatives were interviewed. Participants expressed a strong need for understanding, exchanging information and finding ways to fill their days. They appreciated IPSO's non-obligatory approach and highlight the value of sharing stories with others. Many participants experienced mixed peer groups including both people facing curable and incurable cancer, as uncomfortable. They emphasized the benefit of having separate peer groups for those facing incurable cancer, including able to talk openly about end of life and a deeper sense of understanding and connection. A few participants found these groups inaccessible due to a specific setup. Visitors also experience unpredictability and uncertainty from their illness, making planning difficult, which also affect their IPSO visit.

**Data availability statement:** Data cannot be shared publicly because the respondents did not give informed consent to share the transcripts publicly. This Informed Consent was assessed by the Institutional Review Board METC Brabant (approval number: NW2023-20). Data access requests may be directed to IKNL (info@iknl.nl, https://www.iknl.nl). The complete anonymized transcripts may be made available upon reasonable request and subject to ethical approval and data sharing agreements.

**Funding:** This study was financially supported by the Dutch Cancer Society (project 15139). There was no additional external funding received for this study.

**Competing interests:** The authors have declared that no competing interests exist.

## Conclusion

IPSO centers play a crucial role in supporting people facing incurable cancer. To best meet their needs, it is essential to facilitate peer contact by organizing special peer groups or by introducing those facing incurable cancer to each other in another way. It is important to take into account accessibility. Further research should focus on individuals who need IPSO's informal care but are not able to benefit from it due to their burden of disease.

## Introduction

The incidence of cancer is increasing in the Netherlands, leading to an increasing number of people living with or after cancer. This trend illustrates the profound impact of cancer on patients, families, and healthcare systems. Also the number of patients diagnosed with incurable cancer is increasing, as one out of five cancer diagnoses is classified as metastatic disease [1]. This growing number of patients with metastatic cancer poses a considerable challenge for patients and their relatives [2–3] and for health care systems, underpinnning the essential role of informal care and self-management support. These developments highlights the growing societal and healthcare burden associated with cancer, particularly metastatic disease.

This underscores the need to address the specific challenges faced by patients with metastatic cancer. Patients with metastatic cancer are often facing incurable cancer and therefore in need of palliative care, which shifts the focus of treatment from curative intent towards managing symptoms and optimizing the quality of life. This approach consists of physical, emotional, social, and spiritual aspects [4]. Timely integration of palliative care is essential to address the complex needs of these patients [5–6].

Although palliative care addresses a wide range of needs, research shows that people facing incurable cancer still face unique social challenges, such as loneliness in the face of limited life expectancy [7] and a greater need for information about supportive care and end-of-life planning [2]. These findings suggest that the needs of people facing incurable cancer may differ from those of people facing curable cancer.

In order to meet these needs, support from both healthcare professionals and volunteers is essential. Volunteers are important in supporting patients with incurable cancer, as they can meet the multifaceted needs of patients and their families by providing support for the physical, psychological, social and spiritual aspects of the illness [8].

An important organization providing such volunteer support in the Netherlands is IPSO, IPSO centers for life with and after cancer. This national organization consists of 100 drop-in centers, all dedicated to providing volunteer-led support for people dealing with cancer [9]. These centers are either standalone facilities or located in proximity to hospitals. They offer free, psych-social, non-medical support provided by trained volunteers and professionals, aiming to improve quality of life and reduce isolation during and after cancer treatment. The support offered by IPSO include

in-person support by volunteers (a listening ear, practical support), peer support, and workshops (creative, physical). The volunteers are trained and supervised by a professional coordinator. The services are open for everyone confronted with cancer as a patient, relative or bereaved relative. Visitors report that they visit IPSO centers primarily to relax, but also to seek professional advice and meet peers [10]. IPSO centers provide a unique and accessible form of informal care that addresses both practical and emotional needs of people affected by cancer.

The positive experiences of visitors highlight the value of these centers for people living with cancer [11]; and almost all visitors report that the IPSO centers meet their expectations and needs [10]. The study by Sinzer [11] showed that the vast majority of IPSO visitors experience a positive change on their wellbeing, particularly in relation to their mental well-being and social participation. Moreover, visitors often come multiple times to IPSO, one third visits once a month and one third visits IPSO longer than 3 years [11]. Furthermore, a recent research report showed that the societal benefit of IPSO is equivalent to 150000–200000 consultations with healthcare professionals annually [12]. These outcomes underline the potential of IPSO centers to contribute meaningfully to the wellbeing of patients with cancer and their families.

Despite these positive findings, little is known about how people with metastatic cancer experience this support. A better understanding of these is essential in order to optimize the services provided by IPSO centers. Therefore, this study aimed to assess the experiences and needs of visitors dealing with metastatic cancer concerning the supportive care of volunteers and peers within IPSO centers. This study aims to enhance understanding of the specific needs of patients and relatives faced with incurable cancer.

## Method

### Study design

A qualitative study grounded in a constructivist paradigm using semi-structured interviews was conducted to explore how IPSO visitors facing incurable cancer construct meaning from their experiences within the IPSO centers. For reporting, the COnsolidated criteria for REporting Qualitative research (COREQ) was used (S1 File). This study was exempted from full medical ethical review by the institutional review board METC Brabant (number: NW2023−20) in accordance with the Dutch Medical Research Involving Human Subjects Act.

### Participants and setting

Participants were recruited between August 2023 and December 2023 using convenience sampling until data saturation was reached. Initially, the researcher approached the coordinators of various IPSO centers. These coordinators serve as trusted contacts within the IPSO centers and maintain low-threshold contact with the IPSO visitors. Subsequently, flyers were distributed for recruitment purposes within the center. Following this, coordinators were contacted by telephone, and the researcher visited IPSO centers for a brief introduction to the centers and their volunteers. The coordinators of the IPSO centers then approached visitors. When a visitor expressed interest in participating, they received an information letter from the research team, including an informed consent form. Interested visitors could then contact the research team. The researcher contacted the participants prior to the interview via telephone in order to further explain the study and, if desired, to schedule an appointment for the interview. All participants provided written consent for participating in the interviews. Visitors were eligible to participate if they were facing incurable cancer as a patient, a relative or a bereaved relative; were visitors of an IPSO center, were aged over 18 years, and spoke Dutch. These criteria were all self-reported and not crosschecked with the medical files.

### Data collection

The interview guide was based on concepts emerging from the literature (including care needs on the four dimensions of palliative care, and peer support) and was developed in collaboration with an expert group from various organizations

dedicated to palliative care (IPSO, Agora, VPTZ, and IKNL). The interview guide explored the participants' experiences with IPSO, and their needs for informal care, subdivided into the four dimensions of palliative care (physical, emotional, social, and spiritual). The interview guide included various prompts and sub-questions per domain to promote more in-depth conversations (S 2 File). Participants were also asked to reflect on their experiences with IPSO and potential areas of improvement. A test interview was performed by JvK to evaluate the interview guide with a senior researcher with expertise in qualitative research in order to test the flow of the interview guide and to give feedback to JvK on her interview skills. Subsequently, the interview guide was adjusted based on the results of this test interview.

The interviews were conducted face-to-face between August 2023 and December 2023 (lasting approximately 60 minutes). The interviews took place either at the IPSO center that the participant visited or at the participant's home, ensuring a comfortable and convenient environment for the participant.

All interviews were conducted by JvK (MSc), a female junior researcher with experience in interviewing and qualitative research. The researcher works for the Dutch Comprehensive Cancer Organisation, which is not associated with or related to IPSO centers. Therefore, the independency of the researcher was warranted. No extensive field notes were made during the interviews. Before the start of the interview, sociodemographic information was collected and the participants self-reported about their age, gender, marital status, level of education (using the International Standard Classification of Education) and working situation. The interviews were audio-recorded and transcribed verbatim. All participants received a gift voucher of 20 euros after the interview as an acknowledgement of their participation in the study.

## Data analysis

Analysis of the interviews began during the data collection to enable the researcher to adjust the topic guide where needed (constant comparison) and to determine whether data saturation was reached. Data saturation was based on the description by Morse [13]. Data saturation occurs when categories are theoretically rich and internally coherent. To ensure data saturation in this study, we analyzed the interviews iteratively, comparing emerging themes across participants and refining conceptual categories until no new themes emerged. A total of 20 interviews with patients and 20 interviews with relatives was expected to reach data saturation. This number aligns with established guidelines for qualitative research saturation.

All interviews were coded through thematic analysis, a qualitative analysis technique suitable for exploring themes and patterns within the data. We used the following 6 steps: (1) familiarization with the data, (2) generating initial codes, (3) searching for themes, (4) reviewing themes, and (5) defining and naming themes, and (6) producing the report. The analysis process aimed to identify recurring themes and key patterns related to the research objectives. We conducted an inductive analysis, allowing themes to emerge from the data rather than applying a pre-existing coding frame. We focused on the semantic level of analysis, interpreting the explicit content of the interviews.

Two researchers (JvK and LB) independently coded three transcripts. These initial codes were discussed and compared until consensus was reached. Remaining transcripts were coded by JvK after which the researchers JvK, LB and NR discussed the main (sub)themes, derived from the data. Subsequently, another researcher (SS) also coded more than half of all transcripts (n = 18) and discussed this coding with LB, NR and MB until consensus was reached on the final main themes and subthemes. Participants were not asked to provide feedback on the findings. The software package Atlas.ti 24.2.1 for qualitative research analysis was used for data analysis.

## Results

A total of 31 participants were recruited from 19 different IPSO centers, ranging from 1 to 6 participants per center. The participants 'age (18 patients and 13 relatives and bereaved relatives) ranged from 20 to 80 years old. Most participants (52%) were ranged between 40 and 65 years old (Table 1).

A total of 8 themes and two subthemes belonging to either "experiences" or "needs" were identified (Table 2).

**Table 1. Sociodemographic characteristics of visitors facing incurable cancer of IPSO centers.**

| | IPSO visitors – patients | IPSO visitors – relatives and bereaved relatives |
|---|---|---|
| | (n = 18)<br>N (%) | (n = 13)<br>N (%) |
| **Gender** | | |
| Male | 5 (28%) | 4 (31%) |
| Female | 13 (72%) | 9 (69%) |
| **Age (years)** | | |
| 20-40 | 5 (28%) | 1 (8%) |
| 40-65 | 10 (56%) | 6 (46%) |
| 65-80 | 3 (17%) | 6 (46%) |
| **Marital status** | | |
| Married | 17 (94%) | 6 (46%) |
| Not married | 1 (6%) | 1 (8%) |
| Widow | 0 (0%) | 6 (46%) |
| **Level of education**[1] | | |
| Low | 14 (78%) | 5 (38%) |
| Medium | 0 (0%) | 6 (46%) |
| High | 4 (22%) | 2 (15%) |
| **Working situation** | | |
| Employed | 2 (11%) | 4 (31%) |
| Unemployed | 2 (11%) | 2 (15%) |
| Incapacitated | 10 (56%) | 1 (8%) |
| Retired | 3 (17%) | 5 (38%) |
| Volunteer | 1 (6%) | 1 (8%) |

1) Education was self-reported by the participants and defined according to the guidelines of the International Standard Classification of Education as low (no education, pre-primary education, primary education, lower secondary education, compulsory education, initial vocational education), medium (upper secondary general education, basic vocational education, secondary vocational education, post-secondary education) and high (specialized vocational education, university/college education, (post)doctorate and equivalent degrees).

**Table 2. Overview categories, themes and subthemes.**

| Category | Theme | Sub theme |
|---|---|---|
| Experiences of visitors dealing with incurable cancer with IPSO centers | • Sharing stories and experiencing recognition<br>• A warm place without requirements<br>• Feeling out of place among peers who are not facing incurable cancer<br>• Special peer groups with people facing incurable cancer | ◦ A valuable concept<br>◦ The set-up: not always easy to access |
| Needs of visitors facing incurable cancer | • A need for understanding<br>• A need for information<br>• A need to fill the day<br>• A need to visit while not being able to | |

## Experiences of visitors facing incurable cancer with IPSO centers

The following four themes and two sub themes relate to the experiences of participants within IPSO centers. The experiences of patients, relatives and bereaved relatives are not described separately, but are combined together.

### Sharing stories and experiencing recognition

Many participants reported that they feel comfortable sharing their stories in IPSO centers, which is one of the reasons why they keep coming back. Some preferred to share their stories with peers, others prefer to share them with volunteers of IPSO, sometimes specifically one-on-one. Many participants indicated that they feel heard and experienced recognition.

*"Recognition. And feeling that people understand what you're talking about. But I also really enjoy listening. To hear how someone else experiences and deals with it." (Relative 27)*

When sharing their story, participants experienced a sense of recognition and acknowledgment among others. This helped them to cope with the incurable cancer.

*"And well, since then, that walk-in house has been a great support for my wife, and therefore also for me, to process. My wife, the fact that she was so ill. And for me, just the fact that there is indeed a non-medical institution that still offers warmth and genuine interest and attention. So you come into an environment with people who understand you, who understand my wife." (Relative 29)*

In addition, visiting an IPSO center helped them to put things, like emotions, into perspective or to find something positive, especially when listening to other peoples' stories.

*"When you've spoken here, or been here, you leave it here, and then you can just continue life at home." (Patient 3)*

### A warm place without requirements

Participants indicated that they experienced some barriers to visiting the IPSO center, but once they went, they found the IPSO center to be a warm, pleasant and welcoming place. When they visit the IPSO center, they received warm attention and felt being seen.

*"My neighbor said, 'Come on, let's go together,' which was very nice. And then I was here, and I thought, what have I been worrying about all this time? Because you really come into such a warm bath, and that is so nice." (Patient 16)*

*"You immediately feel very welcome there. Among other things, because they make the effort to remember who you are. It's always like, when I walk in, 'Hey [name], how are you?' Yes, that contributes to the feeling that you are seen." (Patient 9)*

Moreover, many participants appreciated IPSO's open approach, which allowed them to visit when it suits them. In addition, most participants experienced the volunteers of IPSO working as host as approachable and felt that IPSO is a place where they can always turn to.

*"If I need something, like when I come in this morning and can tell my story, then I can get it off my chest. And I don't always have to, but if I want to then I can." (Patient 1)*

### Feeling out of place among peers who are not facing incurable cancer

Some participants participated in mixed peer groups, consisting of visitors with cancer from all stages of the disease, ranging from recently diagnosed, under treatment, cured and incurably ill. Most of them expressed that they often feel out of place in such mixed peer groups. They felt they could not speak freely in these mixed groups, because they were afraid of burdening others with their story of terminal illness or taking away hope from those who had a chance of recovery.

*"And that is my own barrier [...] that I often found it difficult, because it is very clear that I will not get better. And with that, I felt a bit like I also represent the fear of many other people who are there. Not everyone who comes there doesn't get better. [...] And that's why I find it difficult to really say what is bothering me." (Patient 18)*

*"And what I also said to you, yes, I would really weigh my words carefully because I want to handle that cautiously." (Patient 7)*

Some participants experienced it as painful to participate in a mixed peer group, as it was confronting for them to hear that people with curable cancer were returning to their original lives again, which was not the case for them.

*"The group I first joined, I'm considering leaving it. Because I notice that they continue with their lives, and that's very understandable. And I think... yes, that's difficult." (Patient 3)*

*"Because they talk about how to deal with some complaints while they are better, or getting better in their lives. And we have to learn to live with it, but yes, our final destination is, of course, death, and hopefully not for them for a long time. So you have different conversations." (Patient 2)*

**Special peer groups with people facing incurable cancer**

In some IPSO centers, special peer groups are organized specifically for visitors facing incurable cancer and sometimes even specifically for patients facing incurable cancer, for relatives or bereaved relatives.

• *A valuable concept*

These special peer groups for people facing incurable cancer are generally perceived as very valuable by the participants. They experienced it as a unique setting where they can talk openly and freely about their situation, including end of life, energy management and time allocation.

*"You understand each other on a completely different level because you share that with each other. For me, it is a very safe place because I can talk about the fact that I have to deal with my mortality there [...] I really enjoy sitting at a table with people and exchanging thoughts who are also in that situation. [...] It is, in any case, a topic that you can hardly discuss anywhere else, in my opinion." (Patient 9)*

Participants reported that it is very easy to connect with people who are dealing with the same situation. Some participants explicitly reported a deep sense of understanding and connection in these groups and many experienced recognition in their peers.

*"It's really different when you talk to people who are in the same boat. And also to talk about wishes or desires or things you just can't do anymore, it seems like it's easier to mention those things." (Patient 18)*

In some IPSO-centers special peer groups for relatives or bereaved relatives are organized. Participants reported the same positive experiences about these groups.

*"And when it's also with those who are bereaved, it feels very familiar. A woman who also had a husband who died of pancreatic cancer [...]. So I think, yes... that's just pleasant. She knows what it is like." (Bereaved relative 24)*

Not all participants expressed having a need for specific peer group for visitors facing incurable cancer for example, because they believed the focus here is on the end of life.

- *The set-up: not always easy to access*

IPSO centers organize special peer groups for people with palliative cancer in different set-ups. Some peer groups take place in the form of a walking group, which is experienced in various ways. Although some participants reported to enjoy the set-up of a walking group, some other participants reported that the distinct set-up resulted in them being excluded from participating. These participants could not participate due to physical limitations and were not able to join a special peer group for people facing incurable cancer, as this group was the only existing special peer group in their IPSO center.

*"I think it's wonderful, but when I read 'in [name of nature reserve],' I thought, well, I don't think I can manage that, you know. And they walk an average of 5 kilometers." (Patient 10)*

**Needs of visitors facing incurable cancer**

The following four themes relate to participants' needs, in the context of IPSO. The experiences of patients and relatives are, again, not described separately, but are combined together.

**A need for understanding**

Many participants reported that they find it difficult to talk about their illness and situation with their family and friends. Some participants reported feeling frustrated by how their social environment talks to them about their illness. According to participants, their illness as a topic is avoided by many or overemphasized by others. In addition, participants indicated that their environment sometimes misinterprets them when they talk about how they are doing. Participants express to find this frustrating.

Others reported being able to openly discuss their illness with loved ones and receiving a lot of love and support from them. However, they still expressed a need for real understanding as no one around them truly understands what it is like to live with the knowledge that they are incurable.

*"In the beginning, I really struggled with the fact that when you talk to friends or family, or at home with my wife or the children, for example, everyone understands the situation to a certain extent, but no one understands 100%, let's say, what it's like to hear that you won't get better. Everyone can imagine it to a certain extent. And at some point, I ran into that, that you can talk to people." (Patient 8)*

Some participants also expressed a need for understanding and attention to the personal impact of the disease, as these aspects were not addressed in the hospital.

*"There is rarely anything asked about the psychosocial side, or the experience of the disease, which is really a secondary thing." (Patient 10)*

**A need for information**

Participants have different information needs, including information about the disease and specific treatments and information about what to expect in terms of the course of the disease and the end of life. Participants also expressed the need for practical information about what they need to arrange when approaching the end of life (e.g., advance care planning, financially).

*"Of course, palliative care does have its own specific questions, you could say." (Patient 3)*

Finally, they appear to need a point of contact who can advise them on which authority to contact on which question. The latter is particularly true for participants who find themselves in a palliative stage for some time and whose health situation is quite stable. They experienced an information gap because they are no longer treated in the hospital.

Participants gather this information in different ways, some reported they exchange tips and information during peer group sessions, while others attended IPSO information evenings focused on palliative care. Although some participants indicated that they were sometimes unsure whether IPSO was the right place for their questions, they were always listened to and the IPSO volunteers were always willing to engage thoughtfully with their concerns, which made them feel satisfied.

*"Exactly, so now that I'm at [name of IPSO center] and have a good feeling about it, I also know that if I need something, I can turn to them, and if they not be able to help me, they know where I should be then, and that gives peace of mind, and that's actually enough." (Relative 27)*

**Finding ways to fill the day**

Participants who are quite stable in the palliative phase, expressed a clear need to find meaningful ways to fill their days, as they no longer spend much time in medical settings. This challenge is an issue that concerns quite some participants, especially for those who had to give up their job because of their illness with little likelihood of returning. With longer life expectancies and relatively high energy levels compared to those with more rapidly progressing conditions, these participants feel a clear need for purposeful activities. Many participants fulfilled this need by making visits to IPSO.

*(In response to the question of what this participant hoped to find at IPSO)*

*"'A bit of daytime activity, because I have quite an extensive network, you know, I do a lot, still in the evenings and on weekends. But during the weekdays, during the day, all my friends are working." (Patient 4)*

**A need to visit while not being able to**

Participants indicated sometimes being unable to visit IPSO even though they have a need for it. Some participants said this was because their illness, or of their loved one, requires a lot of their time, due to frequent hospital visits or the amount of care taking. Some relatives indicated they therefore did not have time to visit IPSO at all. They started to visit IPSO after the death of their loved one.

*"No, but this is of course something quite recent that things are going well again. If you had come a month ago, I was really out of it, I couldn't do anything." (Patient 18)*

*"And when my husband was almost dying, I did not go anymore because I could only be at home. But when he passed away, [name IPSO host] asked if I still wanted to talk. I think that's actually just as important, that you are still heard and can talk afterwards. Because during the time he was sick, you are only busy with caring, and I had really made it my goal to make his quality of life as good and comfortable as possible. And that worked, I'm very happy about that. But afterwards, it's your turn a bit." (Relative 28)*

Some participants were confronted with an unpredictable course of their illness, or experienced many ups and downs physically. This makes it difficult to plan a visit in advance, as one often doesn't know one's condition. According to participants it sometimes occurred they don't visit IPSO for times because they are too weak to do so.

*"What is most present now is the uncertainty. That is very specific to this phase. Not this weekend, but the weekend after, I have booked a city trip to Berlin, taking my mobility scooter, things like that. [...] But it might not happen at all,*

*you know. And I don't know. Is this still 2 months, is this half a year? And that is, of course, very different from being in the middle of the treatment phase."* (Patient 18)

*"Yes, I have had really bad times, where I was just not able to go."* (Patient 18)

Moreover, some participants reported having concerns about how to continue attending IPSO in the future as their disease progresses.

*"Because sometimes I think, how will I get here if I can't drive anymore? [...] If you use morphine, you are not allowed to drive. Then you start thinking, if it becomes permanent in the future, how will you still get here, because you want to keep coming."* (Patient 7)

## Discussion

### Main results

This study explores the experiences and needs of visitors of IPSO Centers for life with and after cancer, who are facing incurable cancer. Regarding the four dimensions of palliative care, which are psychological, physical, social and existential, we found that visitors of IPSO facing incurable cancer have specific needs mainly in relation to the psychological and social dimensions. Besides that, this group experienced IPSO centers as very positive and of great value in coping with their incurable disease, especially through connecting with peers. The participants also reported the added value of dedicated peer groups for patients or relatives facing incurable cancer, as they felt able to talk openly about end of life and experienced a deeper sense of understanding and connection. However, the volatility and unpredictability of the disease makes that they cannot always visit IPSO when they have a need to and hampers the practical aspects of organizing dedicated peer groups.

A finding that deserves particular attention relates to dedicated peer support for patients and relatives facing incurable cancer. Notably many visitors facing incurable cancer feel out of place in peer groups that also include visitors dealing with curable cancer. In addition, attending dedicated palliative peer groups are experienced as very positive, as also described by Li et al. [14]. Research on peer support for patients with incurable or advanced cancer is scarce, as shown by Walshe et al. in a scoping review [15]. However, a recent feasibility study in the UK showed promising results of dedicated peer support for patients with advanced cancer. The recruitment of peer mentors (with advanced cancer) was unproblematic, the recruitment of participants was challenging, and this feasibility study showed promising results, indicating a positive effect on QoL [16] These findings align with our visitors' need to be understood and are mutually reinforcing.

Visitors report that their need to visit an IPSO center sometimes cannot be met due to the time demands of their illness or because the course of the illness involves a lot of uncertainty and volatility. "The high symptom burden in people with incurable cancer has been known for some time [17], but the uncertainty is a relatively recent development in incurable cancer [18]. Geijteman et al. [18] conclude that patients facing incurable cancer are living longer because of better treatment options, but in greater uncertainty. They also point out the physical, psychological, social and spiritual challenges this development brings to patients [18]. This development shows that the variety in needs within the group of people facing incurable cancer increases. This requires a more flexible approach regarding the services of IPSO.

### Implications and further research

This study highlights the importance of tailoring psychosocial support for patients and families facing incurable cancer. The findings underscore the need for dedicated peer groups that address the unique emotional and existential challenges, as mixed groups with curable cancer patients may inadvertently hinder open dialogue. Finally, for volunteers, it is

important to be well informed about the needs and experiences of visitors facing incurable cancer, including advance care planning, end-of-life care, and bereavement.

A key recommendation for future research is to investigate how many individuals facing incurable cancer may need informal care but are unable to access it due to their substantial burden of illness. It is important to explore the extent to which these illness-related limitations affect access to informal support in order to develop appropriate tailored care for this group. Future research should also explore flexible and accessible models of support that respond to evolving needs and symptom burdens, while also reflecting the cultural diversity of participants and their potentially cultural-specific needs. While many current visitors have a Western European background, it is important to examine how IPSO can optimize its services in terms of accessibility and culturally suitability for individuals from non-Western European backgrounds. Additionally, investigating digital or hybrid formats for peer support may offer promising avenues to overcome practical barriers related to physical attendance and timing.

By exploring in depth these aspects in future research, IPSO centers could better address the needs of its visitors in the palliative phase and optimize their support structures, including digital or hybrid formats for peer support. This will not only improve the experience of current visitors, but also help visitors who may not return or don't visit at all to receive the informal supportive care they need.

## Strengths and limitations

A limitation of the study is that as the study focused on visitors, their experiences will be predominantly positive; otherwise, they would not continue to visit. In addition, as the coordinator was free to choose who he/she approached to participate in the study, this may also lead to a positive bias. This limits the richness of the findings.

A strength of the study is the diversity of the participants. Participants were recruited from many different IPSO centers and since IPSO centers differ in offering various activities and may also be organized somewhat differently, this provides a broad view of the experiences and needs of IPSO visitors facing incurable cancer. As IPSO centers vary considerable in size and support, this study contains a great richness in experiences and needs of IPSO visitors facing incurable cancer. Due to the lack of contextual information about participants, such as their ethnicity and specific cancer type, it is difficult to assess whether this may have influenced their experience at IPSO.

## Conclusion

IPSO centers play a crucial role in supporting people facing incurable cancer. It is essential to facilitate peer contact by organizing dedicated peer groups or by introducing those facing incurable cancer to each other in another way, as their need to be understood, their need for information, and their need to fill the day may be addressed through peer contact. When organizing specific activities for visitors facing incurable cancer, it can be recommended to take into account flexibility and accessibility, considering the unpredictable and volatile course of their disease.

## Supporting information

**S1 File. This is the COREQ checklist.**
(DOCX)

**S2 File. This is the interview guide.**
(DOCX)

## Acknowledgments

We would like to thank all patients and relatives who participated in the interview study.

## Author contributions

**Conceptualization:** Suzanne D. Siemelink, Julia van Koeveringe, Saskia van Veen, Sandrina Sangers, Natasja Raijmakers.

**Data curation:** Suzanne D. Siemelink, Julia van Koeveringe, Monique Bussmann.

**Formal analysis:** Suzanne D. Siemelink, Julia van Koeveringe, Monique Bussmann.

**Funding acquisition:** Saskia van Veen, Sandrina Sangers, Natasja Raijmakers.

**Project administration:** Saskia van Veen, Natasja Raijmakers.

**Supervision:** Monique Bussmann, Natasja Raijmakers.

**Writing – original draft:** Suzanne D. Siemelink.

**Writing – review & editing:** Suzanne D. Siemelink, Julia van Koeveringe, Monique Bussmann, Saskia van Veen, Sandrina Sangers, Natasja Raijmakers.

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
