## [Decision Letter · Decision Letter 0]

19 Aug 2025

Dear Dr. Raijmakers,

Thank you for submitting your manuscript to PLOS ONE. After careful consideration, we feel that it has merit but does not fully meet PLOS ONE’s publication criteria as it currently stands. Therefore, we invite you to submit a revised version of the manuscript that addresses the points raised during the review process.

**ACADEMIC EDITOR: ** Dear Authors, could you please address the feedback suggested by the two reviewers. Additionally, in the abstract it states ‘in order to realise an appropriate fit at IPSO for this group’ is unclear. Could this please be amended? Perhaps ‘to optimise the services provided by IPSO centers’ as per the introduction. Additionally, the COREQ guidelines are missing for this submission as well as the interview guide as supplementary material

We look forward to receiving your revised manuscript.

Kind regards,

Gursharan K Singh, PhD

Academic Editor

PLOS ONE

Journal Requirements:

[This study was financially supported by the Dutch Cancer Society (project 15139)].

3. Thank you for stating the following in your manuscript:

[This study was financially supported by the Dutch Cancer Society (project 15139).]

[This study was financially supported by the Dutch Cancer Society (project 15139)]

5. Please amend your list of authors on the manuscript to ensure that each author is linked to an affiliation. Authors’ affiliations should reflect the institution where the work was done (if authors moved subsequently, you can also list the new affiliation stating “current affiliation:….” as necessary).’

Reviewers' comments:

Reviewer's Responses to Questions

**Comments to the Author**

1. Is the manuscript technically sound, and do the data support the conclusions?

Reviewer #1: Yes

Reviewer #2: Yes

2. Has the statistical analysis been performed appropriately and rigorously?

Reviewer #1: N/A

Reviewer #2: N/A

3. Have the authors made all data underlying the findings in their manuscript fully available?

Reviewer #1: No

Reviewer #2: No

4. Is the manuscript presented in an intelligible fashion and written in standard English?

Reviewer #1: Yes

Reviewer #2: Yes

Reviewer #1: Thank you for the opportunity to review the paper titled “Experiences and needs of patients with incurable cancer and their relatives with informal care in psychosocial supporting centres in the Netherlands: a qualitative study”. This paper describes the experiences and needs of people impacted by incurable cancer who visited an IPSO centre in The Netherlands for supportive care. The paper provides some insightful findings regarding the value and suitability of these services for this population group. Please see my detailed comments below.

Criteria:

1. The study presents the results of original research. Yes.

2. Results reported have not been published elsewhere. Yes.

3. Experiments, statistics, and other analyses are performed to a high technical standard and are described in sufficient detail. Yes.

4. Conclusions are presented in an appropriate fashion and are supported by the data. Please see my suggested edits below.

5. The article is presented in an intelligible fashion and is written in standard English. Predominantly. Please see minor suggestions below.

6. The research meets all applicable standards for the ethics of experimentation and research integrity. Yes.

7. The article adheres to appropriate reporting guidelines and community standards for data availability. Unsure - I am unable to see the completed COREQ checklist.

Abstract:

• Line 12, page 2 – this sentence would be better suited to your Results section.

• If word count permits, please briefly elaborate on the perceived benefits of having separate peer groups for those with incurable cancer, and what aspects of the group set-ups made them inaccessible for some participants.

Introduction:

• Overall, the introduction is well-written and provides a clear and concise overview of what is currently known on this topic and justification for the aims of this study.

• I would recommend reviewing the paragraph structure in the introduction to improve readability. Ensure each paragraph includes a clear topic and concluding or linking sentence.

• Please provide a short explanation of what is meant by ‘informal caregiver’ in this study and expand on the justification as to why it is important to consider the perspectives of this population group, particularly in the context of incurable cancer.

• Line 22, page 3 – if possible, please define the acronym ‘IPSO’. I would also suggest tweaking the wording slightly to ensure it’s clear that this organisation offers ‘volunteer-led’ support, rather than support for volunteers per se. In addition, could you elaborate further on the reach of this organisation? For example, how widely distributed are the centres? Do they offer online/telephone support in addition to in-person support? Are services accessible free of charge? What are their opening days/hours? Are services general or targeted to specific groups including those impacted by incurable cancer?)

• Lines 1-2, page 4 – ensure consistency in the punctuation of ‘wellbeing’.

• Line 3, page 4 – ‘impact’ on what? Please elaborate.

• Lines 4, page 4 – are decimal places on these numbers necessary?

Methods:

• The authors stated that they used the COREQ checklist for their reporting; however, I cannot see this. Could the authors please confirm if this was submitted with their manuscript?

• Could the authors please elaborate on how purposive sampling was used? For example, did they purposively recruit to ensure variation in age, gender, and other demographic characteristics, or focus on people with specific experiences with the IPSO service? Otherwise, I wonder if this is more convenience sampling.

• How many IPSO centres were contacted and then subsequently agreed to distribute recruitment flyers to their visitors?

• How was study eligibility for each visitor assessed?

• I commend the authors for collaboratively developing an interview guide based on a review of the literature and expert opinion, and for piloting this interview guide before use. I encourage the authors to include this guide in their Supplementary Materials, and to provide more detail on the following: i) what concepts were identified in the literature review and subsequently applied to the interview guide? ii) who was the interview piloted with?

• The authors have indicated that ‘data saturation’ was defined based on the description by Morse (1995). To support reader understanding, could the authors please elaborate on how this definition was applied in the context of their study?

• Please provide a citation for the data analysis method used. It seems to reflect more of a thematic analysis approach rather than a content analysis approach.

• Demographic characteristics of participants are reported. Please clarify in Methods how these data were collected.

Results:

• Table 1 - I am not sure I understand the footnote about level of education. Please provide further explanation. I have also not seen education categorised as ‘low’, ‘middle’, and ‘high’ before – for an international audience, please define what these represent.

• Line 2, page 8 – I do not think it is necessary to repeat the sample size – either report it in the Methods or Results. If possible, I encourage the authors to include a participant flow diagram showing the number of i) visitors to the IPSO centres and/or visitors approached during recruitment, ii) visitors screened as eligible, iii) visitors who consented to participate, and iv) visitors who completed an interview, to improve transparency around recruitment and sampling. As per the COREQ checklist, this should also include reasons for not participating.

• Line 3, page 9 – ‘IPSO hosts’ is a new term here that has not yet been introduced. Please clarify what is meant by this.

• Lines 30-31, page 11 – could the authors please elaborate on this finding by explaining why some people did not perceive it necessary (or ideal) to have specific peer groups for people impacted by incurable cancer?

• Line 21-23, page 12 – “Participants indicated that their environment sometimes draws the wrong conclusions when they talk about how they are doing, which they find frustrating” – I am a little unsure what is meant by this. Could the authors please clarify (or perhaps include an example)?

• Line 8-9, page 13 – the second use of the word ‘practical[ly]’ is somewhat redundant in this sentence.

• I think it would be worthwhile including a quote to illustrate the theme of ‘a need for information’ specifically one that highlights the need for disease-specific and/or practical information.

• Lines 10-11, page 14 – avoid using contraction words – e.g., ‘didn’t’.

Discussion and conclusion:

• Overall, the authors have raised some important points regarding the suitability and value of supportive care services, such as those offered at IPSO centres, for people impacted by incurable cancer, that emerged from their findings. I recommend that the authors elaborate further on their findings regarding peer groups that are exclusive to people diagnosed with incurable cancer or to bereaved relatives. Given that not all participants supported this approach, further discussion of the perceived advantages and disadvantages would strengthen the interpretation of these findings. I also recommend that the authors compare their findings to previous research conducted with IPSO visitors, to demonstrate the novelty and uniqueness of their findings for people impacted by incurable cancer.

• Line 25, page 15 – please check the punctuation here. There is an inverted comma at the beginning of a sentence.

• Line 30, page 15 – remove ‘that’.

• For improved flow, I would suggest including the ‘Implications and further research’ section above ‘Strengths and limitations’

Strengths and limitations:

• To clarify, were all participants returning visitors as opposed to once-off visitors? If so, this should be clarified in Methods.

• Another limitation could be a lack of contextual information for individual participants that could have an impact on their experience with the service (e.g., race/ethnicity, geographic remoteness, cancer type, duration and nature of engagement with services in the IPOC centres).

Implications:

• Lines 13-19, page 16 – perhaps provide some specific examples of how these recommendations could be achieved.

Conclusion:

• Line 6, page 17 – Please soften the language of “[needs]…will be addressed through peer contact”. The findings of this study allude to the fact that special peer groups may be of benefit to some people impacted by incurable cancer, but we cannot say for sure that these groups will address all the aforementioned needs.

Reviewer #2: Data availability:

I completely understand that raw qualitative data is not appropriate to be made available publicly due to ethical/confidentiality concerns; I simply feel this needs to be stated more clearly in the submission.

Please see attached file for all other feedback.

**Do you want your identity to be public for this peer review?** For information about this choice, including consent withdrawal, please see our Privacy Policy

Reviewer #1: No

Reviewer #2: No

---

## [Author Response · Author response to Decision Letter 1]

29 Oct 2025

Response to reviewers:

Reviewer #1:

Thank you for the opportunity to review the paper titled “Experiences and needs of patients with incurable cancer and their relatives with informal care in psychosocial supporting centres in the Netherlands: a qualitative study”. This paper describes the experiences and needs of people impacted by incurable cancer who visited an IPSO centre in The Netherlands for supportive care. The paper provides some insightful findings regarding the value and suitability of these services for this population group. Please see my detailed comments below.

Response: We thank the reviewer for the thorough review and nice words regarding our manuscript.

Abstract

1. Line 12, page 2 – this sentence would be better suited to your Results section.

Response: We thank the reviewer for this comment and have replaced this line to the Results section.

2. Abstract: If word count permits, please briefly elaborate on the perceived benefits of having separate peer groups for those with incurable cancer, and what aspects of the group set-ups made them inaccessible for some participants.

Response: We thank the reviewer for this comment and have added the perceived benefits of a separate peer group to the results, see page 2, line 22 “They emphasized the benefit of having separate peer groups for those facing incurable cancer, including able to talk openly about end of life and a deeper sense of understanding and connection”

Introduction:

3. Overall, the introduction is well-written and provides a clear and concise overview of what is currently known on this topic and justification for the aims of this study.

Response: Thank you

4. I would recommend reviewing the paragraph structure in the introduction to improve readability. Ensure each paragraph includes a clear topic and concluding or linking sentence.

Response: Thank you for this comment, we have revised the paragraph structure of the introduction and have added linking sentences to it, see page 3-4

5. Please provide a short explanation of what is meant by ‘informal caregiver’ in this study and expand on the justification as to why it is important to consider the perspectives of this population group, particularly in the context of incurable cancer.

Response: Thank you for this comment, we understand the confusion. In this context, an informal caregiver is a person who provides care and support to someone with incurable cancer without being paid or professionally trained. They are often family members, friends, neighbours, or other close contacts of the person receiving care. Caregivers who work at IPSO are trained, and therefore are addressed as volunteers in our manuscript. They are important for this population as they can provide extensive psychosocial support, where the time of formal / medical caregivers is limited.

6. Line 22, page 3 – if possible, please define the acronym ‘IPSO’. I would also suggest tweaking the wording slightly to ensure it’s clear that this organisation offers ‘volunteer-led’ support, rather than support for volunteers per se. In addition, could you elaborate further on the reach of this organisation? For example, how widely distributed are the centres? Do they offer online/telephone support in addition to in-person support? Are services accessible free of charge? What are their opening days/hours? Are services general or targeted to specific groups including those impacted by incurable cancer?)

Response: Thank you for this comment, we have added additional information about the IPSO centers in the Netherlands, see page 3. “This national organization consists of 100 drop-in centers, all dedicated to providing volunteer-led support for people dealing with cancer [9]. The support offered by these IPSO include in-person support by volunteers (a listening ear, practical support), peer support, and participation in workshops (creative, physical). The volunteers are trained and supervised by a professional coordinator. Visitors report that they visit IPSO centers primarily to relax, but also to seek professional advice and meet peers [10]. IPSO centers provide a unique and accessible form of informal care that addresses both practical and emotional needs of people affected by cancer.” and page 4: “Most services are open for everyone confronted with cancer as a patient, relative or bereaved relative and not specifically targeted”

7. Lines 1-2, page 4 – ensure consistency in the punctuation of ‘wellbeing’.

Response: We have checked consistency of wellbeing and changed it all to wellbeing.

8. Line 3, page 4 – ‘impact’ on what? Please elaborate.

Response: Thank your for pointing this out. We have changed impact into societal benefits, as this study was on the benefits of IPSO for our society and our healthcare system.

9. Lines 4, page 4 – are decimal places on these numbers necessary?

Response: We agree with the reviewer and have removed the decimals.

Methods:

10. The authors stated that they used the COREQ checklist for their reporting; however, I cannot see this. Could the authors please confirm if this was submitted with their manuscript?

Response: We have added the COREQ checklist to the submission. Unfortunately, this did not go through at the first submission.

11. Could the authors please elaborate on how purposive sampling was used? For example, did they purposively recruit to ensure variation in age, gender, and other demographic characteristics, or focus on people with specific experiences with the IPSO service? Otherwise, I wonder if this is more convenience sampling.

Response: We agree with the reviewer that our sampling was more convenience sampling. We did try to have a variation in the role (patient and relatives), but this was marginal. We have altered this in the Method section.

12. How many IPSO centres were contacted and then subsequently agreed to distribute recruitment flyers to their visitors?

Response: All IPSO centers were contacted and received information about the project, including the recruitment flyer, but not all responded. In total, we recruited patients and relatives in 19 different IPSO centers. We have added this information to the Result section, see page 9” “A total of 31 participants were recruited from 19 different IPSO centers, ranging from 1 to 6 participants per center. The participants ‘age (18 patients and 13 relatives and bereaved relatives) ranged from 20 to 80 years old.”

13. How was study eligibility for each visitor assessed?

Response: All IPSO visitors (patients and relatives) were eligible for participation when facing incurable cancer. Incurable cancer was self-reported by the participants and not checked with their medical file. Furthermore, their age was also self-reported at the beginning of the interview. The exclusion criteria of having severe mental illness or psychological problems was not officially assessed and therefore we have removed it from the manuscript. Furthermore, we have added information about age and diagnosis to the manuscript, see page 5. “These criteria were all self-reported and not crosschecked with the medical files.”

14. I commend the authors for collaboratively developing an interview guide based on a review of the literature and expert opinion, and for piloting this interview guide before use. I encourage the authors to include this guide in their Supplementary Materials, and to provide more detail on the following: i) what concepts were identified in the literature review and subsequently applied to the interview guide? ii) who was the interview piloted with?

Response: Thank you for your kind words. We have added the interview guide to the Supplementary Materials and additional information about the literature review and pilot test, see page 5-6. “The interview guide was based on concepts emerging from the literature (including care needs on the four dimensions of palliative care, and peer support) and was developed in collaboration with an expert group from various organizations dedicated to palliative care (IPSO, Agora, VPTZ, and IKNL).”

15. The authors have indicated that ‘data saturation’ was defined based on the description by Morse (1995). To support reader understanding, could the authors please elaborate on how this definition was applied in the context of their study?

Response: Thank you for this remark. We have added additional information to the manuscript to show how we applied this in our study, see page 6: “Data saturation was based on the description by Morse [13]. Data saturation occurs when categories are theoretically rich and internally coherent. To ensure data saturation in this study, we analysed the interviews iteratively, comparing emerging themes across participants and refining conceptual categories until no new insights were observed.”

16. Please provide a citation for the data analysis method used. It seems to reflect more of a thematic analysis approach rather than a content analysis approach.

Response: Thank you for pointing this out. We totally agree with the reviewer that we used a thematic analysis approach. The word content was wrong for which we apologise and has been removed accordingly.

17. Demographic characteristics of participants are reported. Please clarify in Methods how these data were collected.

Response: We have added this information to the manuscript, see page 6 “Before the start of the interview, sociodemographic information was asked and the participants self-reported about their age, gender, marital status, level of education and working situation.”

Results:

18. Table 1 - I am not sure I understand the footnote about level of education. Please provide further explanation. I have also not seen education categorised as ‘low’, ‘middle’, and ‘high’ before – for an international audience, please define what these represent.

Response: Thank you for this comment. We have used the guidelines of the International Standard Classification of Education to define the level of education, based on the self-reported education of the participants. To make this clear, we have added to following information as footnote to Table 1: “Education was self-reported by the participants and defined according to the guidelines of the International Standard Classification of Education as low (no education, pre-primary education, primary education, lower secondary education, compulsory education, initial vocational education), medium (upper secondary general education, basic vocational education, secondary vocational education, post-secondary education) and high (specialized vocational education, university/college education, (post)doctorate and equivalent degrees).|”

19. Line 2, page 8 – I do not think it is necessary to repeat the sample size – either report it in the Methods or Results. If possible, I encourage the authors to include a participant flow diagram showing the number of i) visitors to the IPSO centres and/or visitors approached during recruitment, ii) visitors screened as eligible, iii) visitors who consented to participate, and iv) visitors who completed an interview, to improve transparency around recruitment and sampling. As per the COREQ checklist, this should also include reasons for not participating.

Response: Thank you for your feedback, we have removed the sample size from the Methods, as we believe it is more appropriate in the Results. To improve the clarity of the participants flow, we have added extra information about the number of IPSO centers, and number of participants per center included. Unfortunately, we have no information on the persons who were asked but were not interested in participating. See page 9” “A total of 31 participants were recruited from 19 different IPSO centers, ranging from 1 to 6 participants per center. The participants ‘age (18 patients and 13 relatives and bereaved relatives) ranged from 20 to 80 years old.”

20. Line 3, page 9 – ‘IPSO hosts’ is a new term here that has not yet been introduced. Please clarify what is meant by this.

Response: Thank you for your question. A IPSO host is a volunteer who has the task to welcome all visitors to the IPSO center and to provide them with coffee and tea and information. We have used the term volunteer in the manuscript, to avoid confusion.

21. Lines 30-31, page 11 – could the authors please elaborate on this finding by explaining why some people did not perceive it necessary (or ideal) to have specific peer groups for people impacted by incurable cancer?

Response: Thank you for your remark. Participants did not elaborate a lot about the specific benefits of a mixed group (curable and incurable cancer). However, this is an important nuance to the results, as not all patients expressed the need for a specific peer group as they believe it is only about end of life, and they expressed find benefits of peer support in general, see also Themes Sharing stories and experiencing recognition and a warm place with requirements.

22. Line 21-23, page 12 – “Participants indicated that their environment sometimes draws the wrong conclusions when they talk about how they are doing, which they find frustrating” – I am a little unsure what is meant by this. Could the authors please clarify (or perhaps include an example)?

Response: We have added extra information to this section, see page 13 “In addition, participants indicated that their environment sometimes misinterprets them when they talk about how they are doing. Participants express to find this frustrating.”.

23. Line 8-9, page 13 – the second use of the word ‘practical[ly]’ is somewhat redundant in this sentence.

Response: We agree with the reviewer and have removed this word, see page 14.

24. I think it would be worthwhile including a quote to illustrate the theme of ‘a need for information’ specifically one that highlights the need for disease-specific and/or practical information.

Response: Thank you for this comment. We have added some examples of need for information to the text and included a quote. This quote underlines the phase-specific information needs in general, see page 14

25. Lines 10-11, page 14 – avoid using contraction words – e.g., ‘didn’t’.

Response: Thank you for your feedback, we have altered the contraction words.

Discussion and conclusion:

26. Overall, the authors have raised some important points regarding the suitability and value of supportive care services, such as those offered at IPSO centres, for people impacted by incurable cancer, that emerged from their findings. I recommend that the authors elaborate further on their findings regarding peer groups that are exclusive to people diagnosed with incurable cancer or to bereaved relatives. Given that not all participants supported this approach, further discussion of the perceived advantages and disadvantages would strengthen the interpretation of these findings. I also recommend that the authors compare their findings to previous research conducted with IPSO visitors, to demonstrate the novelty and uniqueness of their findings for people impacted by incurable cancer.

Response: Thank you for this feedback. We have altered the discussion accordingly and believe that the urgency of the need for exclusive peer groups for patients with incurable cancer is better integrated into the Discussion section. See page 16-17 “The participants also reported the added value of dedicated peer groups for patients or relatives facing incurable cancer, as they felt able to talk openly about end of life and experienced a deeper sense of understanding and connection. However, the volatility and unpredictability of the disease makes that they cannot always visit IPSO when they have a need to and hampers the practical aspects of organizing dedicated peer groups.”

27. Line 25, page 15 – please check the punctuation here. There is an inverted comma at the beginning of a sentence.

Response: Thank you, we have altered it accordingly.

28. Line 30, page 15 – remove ‘that’.

Response: That has been removed for this sentence.

29. For improved flow, I would suggest including the ‘Implications and further research’ section above ‘Strengths and limitations’

Response: Thank you for this suggestion, we have added a paragraph on the implications and future research to the Discussion section, see page 16-17: “Implications and future res

---

## [Decision Letter · Decision Letter 1]

10 Dec 2025

Experiences and needs of patients facing incurable cancer and their relatives with informal care in psychosocial supporting centres in the Netherlands: a qualitative study

PONE-D-25-30108R1

Dear Dr. Natasja Raijmakers

We’re pleased to inform you that your manuscript has been judged scientifically suitable for publication and will be formally accepted for publication once it meets all outstanding technical requirements.

Kind regards,

Alexandre Morais Nunes, Ph.D.

Academic Editor

PLOS One

Additional Editor Comments (optional):

Reviewers' comments:

Reviewer's Responses to Questions

**Comments to the Author**

Reviewer #2: All comments have been addressed

Reviewer #3: All comments have been addressed

2. Is the manuscript technically sound, and do the data support the conclusions?

Reviewer #2: Yes

Reviewer #3: Yes

3. Has the statistical analysis been performed appropriately and rigorously?

Reviewer #2: N/A

Reviewer #3: No

4. Have the authors made all data underlying the findings in their manuscript fully available?

Reviewer #2: No

Reviewer #3: Yes

5. Is the manuscript presented in an intelligible fashion and written in standard English?

Reviewer #2: Yes

Reviewer #3: Yes

Reviewer #2: Thank you for addressing my previous comments and congratulations on this paper which I feel addresses an important gap in the literature.

Reviewer #3: The paper presents an interesting and relevant topic. The paper examines the experiences and needs of visitors of IPSO centers (psychosocial support centers for living with and after cancer) for people facing incurable cancer, with the aim of optimizing the services provided by IPSO centers for this group. However, it suffers from several drawbacks that need to be addressed before possible acceptance.

The abstract needs to be rewritten, especially to clarify the methods (one sentence instead of two).

I reviewed the authors’ responses to the editors and compared them with the previous version, and I conclude that the authors carefully implemented all suggested revisions. The previous reviewer was quite demanding, so I believe the article is now in a good position to be accepted.

My only negative point is that the conclusion is very short.

**Do you want your identity to be public for this peer review?** For information about this choice, including consent withdrawal, please see our Privacy Policy

Reviewer #2: **Yes: ** Emma Kemp

Reviewer #3: **Yes: ** Andreia Matos

---

## [Editor Report · Acceptance letter]

PONE-D-25-30108R1

PLOS One

Dear Dr. Raijmakers,

I'm pleased to inform you that your manuscript has been deemed suitable for publication in PLOS One. Congratulations! Your manuscript is now being handed over to our production team.

Kind regards,

on behalf of

Professor Alexandre Morais Nunes

Academic Editor

PLOS One